# Anticancer Activity of Natural and Synthetic Chalcones

**DOI:** 10.3390/ijms222111306

**Published:** 2021-10-20

**Authors:** Teodora Constantinescu, Claudiu N. Lungu

**Affiliations:** 1Department of Chemistry, Faculty of Pharmacy, Iuliu Hatieganu University, 400012 Cluj-Napoca, Romania; 2Department of Surgery, Country Emergency Hospital Braila, 810249 Braila, Romania

**Keywords:** chalcone, azole, cancer, cell line, bioactivity, ligand–receptor interaction

## Abstract

Cancer is a condition caused by many mechanisms (genetic, immune, oxidation, and inflammatory). Anticancer therapy aims to destroy or stop the growth of cancer cells. Resistance to treatment is theleading cause of the inefficiency of current standard therapies. Targeted therapies are the most effective due to the low number of side effects and low resistance. Among the small molecule natural compounds, flavonoids are of particular interest for theidentification of new anticancer agents. Chalcones are precursors to all flavonoids and have many biological activities. The anticancer activity of chalcones is due to the ability of these compounds to act on many targets. Natural chalcones, such as licochalcones, xanthohumol (XN), panduretin (PA), and loncocarpine, have been extensively studied and modulated. Modification of the basic structure of chalcones in order to obtain compounds with superior cytotoxic properties has been performed by modulating the aromatic residues, replacing aromatic residues with heterocycles, and obtaining hybrid molecules. A huge number of chalcone derivatives with residues such as diaryl ether, sulfonamide, and amine have been obtained, their presence being favorable for anticancer activity. Modification of the amino group in the structure of aminochalconesis always favorable for antitumor activity. This is why hybrid molecules of chalcones with different nitrogen hetercycles in the molecule have been obtained. From these, azoles (imidazole, oxazoles, tetrazoles, thiazoles, 1,2,3-triazoles, and 1,2,4-triazoles) are of particular importance for the identification of new anticancer agents.

## 1. Introduction

Cancer is a significant public health problem that has a small number of effective therapies, a poor prognosis, and a high mortality rate [1]. Many cancer cells adapt metabolically to the Warburg effect, which includes increased glucose and nutrient absorption and lactic acid production, even under aerobic conditions. [2] Accurate knowledge of the epidemiology of cancer provides essential information about possible causes and trends of the population for this disease, making possible a favorable intervention to identify effective methods of prevention, monitoring, and diagnosis. [3] The etiology of cancers is influenced by hereditary and environmental factors. For example, altered genetic information has been observed in cancer cells [4]. For this reason, a large number of studies have characterized genomic changes in cancer from oncogenic cell-forming signaling pathways to the spectrum of mutations in different cancer subtypes [5]. Additionally, in oncogenic processes, inflammatory and immune pathways are correlated with numerous cellular and humoral components and have common signaling pathways. In the case of inflammation associated with tumor diseases, processes are long and severe. [6] Inflammation and cancer are known to be correlated in two ways: the intrinsic pathway and the extrinsic pathway. The extrinsic pathway is activated by initiation of oncological processes by inflammation. In the case of the intrinsic pathway, somatic deficiencies and genetic mutations activate signaling pathways and cause an increase in the inflammatory response [7]. Another determinant of cancer is activation of the immune system, which is correlated with many metabolic pathways in cancer cells [8]. In cancer patients, a large number of cells are released into the circulation daily. For the formation of metastases, cancer cells leave the primary site, enter the bloodstream, are subjected to blood vessel pressure, adapt to the secondary cellular environment, and interfere with immune cells [9]. The proliferation of cancer cells is also caused by the accumulation of oxygen species, which have the ability to distort macromolecules and induce cell death [10]. Reactive oxygen and nitrogen species (ROS/RNS) are produced by inflammatory cells and epithelial cells. ROS/RNS cause DNA denaturation in organs under the pressure of the inflammatory process and cause the initiation of carcinogenesis. DNA damage, especially to 8-oxo-7,8-dihydro-2′-deoxyguanosine and 8-nitroguanidine, has been shown to be a molecular mechanism for cancer [11]. Cell apoptosis or programmed cell death is one of the essential methods for regulating carcinogenesis and is a contraction of the cell, which induces DNA fragmentation and chromatin condensation [12,13]. There are two essential apoptotic pathways (death of receptor and mitochondrial pathways). Many studies have identified many potential targets for anticancer therapy [14]. Acting on these targets aims to destroy or stop the growth of cancer cells [15]. Caspases, a group of cysteine proteases that degrade cellular proteins, are important targets for anticancer therapy because they play an essential role in apoptotic signaling [16].The PI3K/AKT pathway is also considered one of the key mechanisms involved in cell migration, invasion, and transition through the pulmonary mesenchymal epithelium. In addition, this signaling pathway is associated with proliferation and metastases in renal cell carcinomas, apoptosis of cells in pharyngeal carcinomas, and influences the progression of cancer cells in the cavity [17].

Rational goal of anticancer therapies is to act on cancer cells without influencing non-tumor cellular components or the tumor microenvironment [18]. Cancer cells formed from normal cells are difficult to treat with conventional chemotherapeutic agents selectively. These agents act through various mechanisms, such as blocking the cell cycle at different stages, inducing apoptosis and preventing the proliferation of cancer cells, and interfering with metabolic reprogramming [19]. Both chemotherapy and radiotherapy induce DNA distortion and cause cell cycle blockage or cell death. However, a new generation of cancer therapies is based on increasing intrinsic tumor cellular effects by incorporating agents with a unique mechanism of action or that have a known intrinsic way of installing resistance to therapy [20].

Cytotoxic drugs are classified, according to their mechanism of action, into alkylating agents, heavy metals (platinum), antimetabolites, cytotoxic antibiotics, and cell cycle blockers. Most cytotoxic compounds act on the integrity of DNA and cell division in cancer cells [21]. Clinical use of platinum complexes as an adjunct in anticancer therapy is based on their ability to cause tumor cell death, as these compounds have a wide range of activities [22]. Reasons for the ineffectiveness of anticancer therapies are metastases, recurrences, heterogeneity, resistance to chemotherapy and radiation, and a decreased immune system capacity. All of these therapeutic failures can be explained by the characteristics of cancer stem cells [23,24,25]. Mesenchymal stem cells are a type of cell commonly used in regenerative medicine. These cells are known to exert suppressive effects on cancer cells [26]. Resistance to therapy continues to be the main limiting factor in the treatment of cancer patients. Current standard therapies (surgery, chemotherapy, and radiotherapy) are deficient due to adverse and toxic effects, patient intolerance, and a low long-term survival rate [27,28,29,30]. Surgical therapy and radiation therapy aim to eradicate localized cancers, and advanced stages of the disease can be controlled only by chemotherapy [31]. In the transport process of a biologically active compound, its diffusion may produce nonspecific interactions, which will lead to a decrease in efficiency and adverse reactions [32]. Among the anticancer therapies, targeted therapies are the most effective because they have a low number of side effects, good viability, low doses are administered, and therapeutic resistance is more difficult to install [33]. For example, nanomedicine is successfully used as a vehicle for the targeted transport of immunostimulatory agents to facilitate an antitumor immune response. Numerous strategies have been investigated to reduce the toxicity of anticancer immunotherapy. Nano-formulations of antigens, cytokines, chemokines, nucleotides, and Toll-like receptor agonists showed favorable results [34]. At present, the identification of new alternative therapeutic agents, which are more effective and have less toxic effects, is attracting growing interest. This goal is challenging to achieve due to the complexity of tumor formations [35]. Monoclonal antibodies and chemoprevention by natural compounds are two important directions for the treatment and prevention of cancer [36]. One of the essential strategies in this regard is the use of biologically active phytochemicals, as they have low toxicity and pleiotropic effects in various cellular processes that interfere with the onset and progression of cancer. Interference with carcinogenesis through the diet or supplementation with natural compounds is called chemoprevention [37,38,39,40,41]. Over 3000 plant compounds with anticancer properties have been identified [42]. Among these compounds, flavonoids have numerous representatives with cytotoxic properties on many types of human cancer cells and are absent of or have diminished adverse effects on normal cells [43]. Flavonoids are polyphenolic compounds and represent a class of biologically active secondary metabolites in plants with a basic structure of diphenylpropane (C6-C3-C6) and that have a low molecular weight. They are biosynthesized from phenylpropanoid and chalcones are the first flavonoids to be formed [44,45,46,47,48,49,50,51]. The common precursor of flavonoids is phenylalanine, and calcium synthetase, calcium isomerase, and flavan 3 hydrolase are considered key enzymes for their biosynthesis [52,53,54,55,56]. For many flavonoids, a bridge forms a pyranic or pyronic ring [57]. Depending on the basic structure, these compounds are classified into chalcones, aurones, flavanones, flavones, isoflavones, dihydroflavonols, flavonols, leucoanthocyanidins, anthocyanidins, and flavan-3-ols (Figure 1) [58,59,60,61].

Structural diversity of these compounds derives from combined effects of flavonoid biosynthesis enzymes with different catalytic and specificity functions [62]. Dietary consumption of flavonoids is associated with a decreased risk of chronic diseases, such as cardiovascular disease, neurodegenerative diseases, asthma, autoimmune diseases, and cancer (especially lung, prostate, stomach, and breast cancers) [63,64,65,66,67,68,69,70,71]. Flavonoids are also known to have many bioactivities, such as anti-allergic, anti-inflammatory, antibacterial, anti-carcinogenic, antioxidant, antidiabetic, antihypertensive, immunomodulatory, hepatoprotective, anti-obesity, hormonal (e.g., estrogen-like activity), and anti-aging properties [72,73,74,75,76,77,78,79,80,81,82,83,84,85]. There are numerous studies showing that flavonoids suppress the growth of tumor cells in vitro and in vivo [86]. Natural small molecule compounds in class of flavonoids are considered to have remarkable physiological effects, have non-mutagenic properties in the human body, and have attracted increasing interest for the identification of new anticancer agents. Anticancer mechanisms of flavonoids include inhibiting cell growth and proliferation by blocking cell cycle, inducing apoptosis and differentiation, or combining these mechanisms [87,88]. In addition, epidemiological studies show that natural flavonoids have a strong antioxidant potential associated with a low incidence of cancer [89,90]. Antioxidant activity of flavonoids is a result of their ability to donate hydrogen atoms from hydroxy groups to free radicals, a mechanism facilitated by extended conjugation conferred by Π electrons from flavonoids [91]. Flavonoids are known to have a significant antioxidant capacity on superoxide anions, hydroxy radicals, and peroxy radicals. Additionally, flavonoids are more effective than ascorbic acid in neutralizing free radicals produced by oxidative stress [92]. In recent years, anticancer activity of flavonoids, in particular their antimetastatic properties, has been recognized and investigated. Their clinical potential in anticancer therapy has been indicated. For example, LFG-500 (C30H32N2O5) is a synthetic flavonoid with anti-inflammatory and anti-cancer properties. This compound also has antimetastatic potential [93].Bioactivities of flavonoids depend on their degree of hydroxylation, structural class, nature and position of existing substituents, conjugations, and degree of polymerization [94]. Many dietary flavonoids are present in a glycosidic form, wherein a saccharide is bound to a phenolic or hydroxy group of the compound [95,96]. Structure of saccharide is a determining factor for bioavailability of flavonoids [97]. Flavonoids are currently essential components of various pharmaceutical, cosmetic, and medicinal formulations [98,99].Low toxicity of these compounds is considered a major advantage of this class [100]. In some cases, glycosylation of flavonoids is responsible for reducing toxic and undesirable effects of these compounds [101].

Chalcones (1,3-diphenyl-2-propen-1-one) are one of the most important classes of flavonoid compounds present in fruits, vegetables, and tea [102] and represent biogenetic precursors of flavonoids and isoflavonoids [103]. They are lipophilic phytochemicals composed of two aromatic residues (an aldehyde and an acetophenone) joined by an α, β-unsaturated carbonyl system of three carbon atoms (Figure 2) [102,104].

α, β-unsaturated carbonyl group is a good Michael acceptor and participates in nucleophilic additions [105]. Chalcones are found in two isomeric forms (*cis* and *trans*), the *trans* form being more thermodynamically stable and, implicitly, the predominant configuration for these compounds (Figure 3) [106,107,108].

Importance of these compounds derives from their simple chemistry, their easy synthesis, and their ability to replace a large number of hydrogen atoms, thus forming a huge number of biologically active derivatives [109]. An important aspect related to chalcones is the possibility of these compounds to easily form carbon–carbon, carbon–sulfur, and carbon–nitrogen bonds, these being precursors for synthesis of various heterocyclic compounds, such as pyrimidines, pyridines, benzodiazepines, pyrazoles, 2-pyrazolines, imidazoles, and all other flavonoids [110,111,112,113,114]. Isomerization of chalcones to corresponding flavanones in presence of acids or bases explains importance of these compounds as ligands (Figure 4) [115]. For example, Pandey et al. obtained 5-nitro-flavanones by refluxing 2-hydroxychalcones in presence of concentrated sulfuric acid [116].

Due to their flexible structure, chalcones can effectively bind to many enzymes and receptors, which explains the many biological applications of these compounds [117]. Another explanation for pharmacological activities of these compounds is conjugation between double bond and carbonyl group present in structure [118]. Bioactivities of chalcones are dependent on the position, number, and nature of substituents on the two aromatic residues (aldehyde and acetophenone). Data from literature show that a huge number of natural and synthetic chalcones have been identified with clinical and pharmaceutical applications, these compounds having anticancer, antibacterial, antiviral, antipyretic, antihypertensive anti-Alzheimer’s, anti-inflammatory, anti-HIV, antioxidant, antiulcer, estrogenic, and neuroprotective activities. Chalcones have ability to inhibit α-glucosidase, MAO-B (monoamine oxidase), tubulin, and tyrosine kinase [118,119,120,121,122,123,124,125,126,127,128,129,130,131,132,133,134,135,136,137]. On the other hand, chalcones, under certain conditions, have oxidizing properties. This effect may be associated with antitumor activity of these compounds and is based on mechanisms such as increased superoxide formation, cellular glutathione depletion, and phenoxide radical generation. Additionally, available studies have demonstrated targeted activity of chalcones on numerous kinases, microtubules, polytherapy-resistant proteins, and various signaling pathways associated with cell survival and death [138]. Interesting structure of these compounds and various biological activities have led to approval of new drugs from chalcone class, such as metochalcone (an anticoleretic drug) and sofalcone (an antiulcer drug) (Figure 5) [139,140].

Data from literature indicate that replacement of aromatic residues of chalcones with heterocycles determines formation of molecules with special biological properties [141].

Hybrid molecules have ability to solve problem of resistance to therapy due to fact that different pharmacophores have multiple mechanisms of action. Because hybridization of molecules is an important method for identifying new therapeutic agents, there are numerous hybrid molecules in clinical trials [142]. For example, introduction of a nitrogen atom favorably modifies basicity of molecules and determines possibility of forming strong bonds with targets. Another important modified property is polarity, which can be used to reduce lipophilic character, causing solubilization in water and favorable oral absorption [143].

It has been observed that biologically active organic molecules with nitrogen in molecule have good anticancer properties. Among molecules with nitrogen, morpholines and piperidines have important activities on different types of cancers [144]. Yadav et al. obtained triazole chalcones with significant anticancer potential on human cell lines [145]. Examples of where introduction of a pharmacophore is favorable for biological activity of compounds are some hydride chalcones with residues of quinazoline, biphenidate, and indole in molecules. Newly formed molecules have ability to determine reversibility of resistance to therapy in case of breast cancers [146]. Nitrogen-substituted benzimidazole chalcones with an alkyl residue or a five- or six-membered heterocycle also have significant cytotoxic effects on breast adenocarcinoma (MCF-7) and ovarian carcinoma (OVCAR-3). Other hydride molecules with cytotoxic activity above standards on human cell lines (MCF-7, MIA-PA-Ca2human pancreatic cancer cells, A549pulmonary adenocarcinoma, HepG2human cancer cell lines) are 1,2,3-triazole chalcones. Hybrid thiazole compounds induce apoptosis by blocking the G2/S phase of the cell cycle and decrease mitochondrial potential on MIA-PA-Ca2 cell lines in pancreatic cancers [147]. Studies of mechanisms of action for 1,2,4-triazole chalcones show that they have ability to induce apoptosis by increasing Bax protein levels, releasing cytochrome C from mitochondria, and activating caspases 3, 8, and 9 [148]. The purpose of this article is to summarize information obtained experimentally and in silico about anticancer activity of some natural and synthetic chalcones.

## 2. Claisen–Schmidt Reaction

Method most widely used to obtain synthetic chalcones is Claisen–Schmidt condensation reaction (Figure 6). This is an aldolization–crotonization reaction between acetophenone derivatives with aromatic aldehydes. Reaction takes place in strongly acidic or basic catalysis under homogeneous conditions [149,150,151,152].

Use of an alkaline medium is more efficient for obtaining chalcones [153]. Claisen–Schmidt condensation in a basic medium involves formation of an acetophenone anion followed by attack of carbonyl group of acetophenone [154]. Reaction proceeds with yields between 10% and 60%. Condensation is performed at 50°C, reaction time being 12–15 h or one week at room temperature [155]. Disadvantages of this method are inability to recover catalyst, formation of secondary compounds, lack of selectivity, long reaction time, extreme reaction conditions, and difficulty of isolating products [156]. New types of heterogeneous catalysts (Lewis acids, Bronsted acids, solid acids, and solid bases) have been identified for synthesis of chalcones with high selectivity. Use of these catalysts avoids side reactions, such as the Cannizaro condensation reaction or Michael addition [157]. Additionally, in order to avoid a disproportionate reaction of aldehyde, an attempt was made to replace it with benzylidene diacetate [155]. Other examples of reactions for obtaining chalcones are the Heck carbonylation coupling reaction, Sonogashira isomerization and coupling reaction, continuous flow deuteration reaction, Suzuki–Myaura coupling reaction, and synthesis reaction mediated by a solid acid catalyst [158,159,160].

## 3. Anticancer Activity

Chalcone derivatives act on various targets, such as aromatase, ATP binding cassette subfamily G member 2 (ABCG2), breast cancer resistant protein (BCRP), activated nuclear B cell growth factor (NF-ĸB), vascular endothelial growth factor (VEGF), and tyrosine kinase receptors (epidermal growth factor receptor (EGFR) and mesenchymal epithelial transition factor (MET), showing important activities in vitro and in vivo in susceptible and therapy-resistant cancers [161,162]. An important mechanism of antiproliferative activity of chalcones is inhibition of tubulin and interference of these compounds with assembly of microtubules, elements that are essential for maintaining shape and function of cells in processes of mitosis and cell replication. Chalcones block cell cycle and induce apoptosis. Presence of a trimethoxyphenyl residue in chalcone molecules is favorable for irreversible antimitotic activity of these compounds, which have ability to interact with cysteine residues in tubulin through a Michael-type addition reaction [163]. Additionally, replacement of trimethoxyphenyl residue in chalcones with a quinoline or quinazoline is favorable for antitubulin activity. These heterocyclic chalcones form hydrogen bonds with rest of Cys241 and strongly bind to the colchicine binding site (similar to combrestatin A-4, Figure 7) [164,165].

### 3.1. Natural Chalcones with Anticancer Properties

#### 3.1.1. Licochalcones (A–D)

Roots and rhizomes of some species of *Glycyrrhiza* are used in traditional medicine for treatment of gastric ulcers, asthma, and inflammation. More than 600 compounds from licorice have been isolated, the main biologically active constituents being saponins and flavonoids. Among flavonoids, a series of retrochalcones, licochalcones A, B, C, D, E, and G, and schinatine were identified. There are numerous studies on biological effects of licorice active compounds, the most important being anti-inflammatory, antimicrobial, antioxidant, antiulcer, cytoprotective, and cytotoxic properties [166,167].

#### 3.1.2. Licochalcone A

Licochalcone A (LA, Appendix A, Compound 1) is a flavonoid isolated from *Glycyrrhiza urakensis*, *G. glabra*, and *G.inflata* (Fabaceae). It has antitumor, anti-inflammatory, antimicrobial, antiparasitic, anti-obesity, antioxidant, and antiosteoporotic properties. Anticancer effect of LA has been demonstrated for different types of cancer cells, including gastric cancer cells BCG-823, HepG2, OVCAR-3 and SK-OV-3 (ovarian cancer cells), MCF-7, and A549 [168,169,170,171]. Studies show that LA induces apoptosis of U87 glioma cells, nasopharyngeal cancer cells, epithelial ovarian carcinoma cells, and bladder cancer cells. Chalcone also has ability to increase autophagy and block cell cycle in breast cancer cells. In addition, it induces apoptosis by suppressing specific protein 1 in breast cancer [172]. LA has low cytotoxicity on embryonic lung fibroblast cells. In addition, it has ability to inhibit tumor growth and attenuate *cis*-platinum-induced toxicity [173]. Mechanisms by which flavonoids act as anticancer agents include inhibition of Akt activity by suppressing hexokinase-2-mediated tumor glycolysis in gastric cancer, downregulation of metalloproteinase 2 expression and induction of apoptosis in oral cancer cells, increased capacity of miR-144-3p to induce stress in endoplasmic reticulum and induce apoptosis in human lung cells, reduced PI3K/Akt/mTor activation and decreased autophagy in breast cancer, blocking G2/M phase of cell cycle, repressing cell invasion by MEK/ERK and ADAM 9 signaling pathways in human glioma cells, and inducing caspase-dependent apoptosis in human liver cells [174]. Another mechanism by which LA exhibits a strong cytotoxic effect is ROS-induced apoptosis. For example, chalcone induces oxidative stress and, consequently, apoptosis of T24 cells (cell lines derived from human urinary bladder carcinoma) through mitochondrial-dependent pathways and by inducing oxidative processes in endoplasmic reticulum [175]. Another study on cytotoxicity and genotoxicity of LA was developed by Bortolotto et al. LA was compared to *trans*-chalcone (Appendix A, compound 2). Cytotoxicity of natural chalcones (LA and *trans*-chalcone) on MCF-7 and 3T3 (embryonic fibroblast cell lines) was determined at 24 and 48 h. Results indicate marked cytotoxic activity after 48 h of treatment. A cytotoxicity MTT (3-[4,5-dimethylthiazol-2-yl]-2,5-diphenyl tetrazolium bromide) assay showed a dose-dependent response on MCF-7 cells treated with *trans*-chalcone and LA. For compounds analyzed, genotoxicity was found to be better on MCF-7 cells compared with 3T3 cells. DNA distortion causes the immune system to activate in order to eliminate destroyed cells. However, activating immune system to reduce degraded DNA cells contributes to a chronic inflammatory process. Additionally, cellular response to degraded DNA can be corrected by inducing intrinsic apoptotic pathway. G1 phase is a state that precedes DNA replication, in which factors such as cellular conditions (metabolism, signaling, and cell size) influence progression of cell cycle, causing DNA to rebuild in cell or initiating process of apoptosis. IC50 of LA is 60.46 μM. *Trans*-chalcone induces cell cycle blockade in G1 phase and intensifies cellular apoptosis. It has been proposed that treatment with LA and *trans*-chalcone induces apoptosis through mitochondrial pathway, considering that induction of crucial genes of this pathway occurs after 24 h, such as protease activator factor 1 of protein apoptosis and protein X associated with Bcl-2. Although MCF-7 cell lines are known to have a caspase 3 deficiency, a study showed that treatment with these chalcones induces cleavage of poly (ADP-ribose) polymerase (PARP), a polymerase whose cleavage into two fragments is an indicator of apoptosis. Repression of Bcl-2 gene by LA and *trans*-chalcone in an experimental PCR analysis showed, at protein level, a dose-dependent administration to MCF-7 cells and an intrinsic pathway mediation. Cyclin D1 is another protein suppressed on MCF-7 cell lines in presence of chalcones. This protein is crucial for the progression from G1 to S phase and is an important biomarker for some cancers, including breast cancer.For these reasons, cyclin D1 degradation is an attractive target for identification of new antitumor agents and is correlated with cell cycle blockade [176].

Qui et al. also examined effect of LA on lung cancer cells in vitro. Flavonoid treatment significantly decreased viability of A549 and H460 cells (human non-small-cell lung carcinoma), this change being strongly influenced by dose. A total of 40 μM of licochalcone suppresses growth of lung cancer cells by 45–80% after 24 or 48 h of treatment. In addition, compound exhibits low cytotoxicity on normal human lung epithelial cells.To highlight whether one of mechanisms of inhibition of lung cancer cell growth by LA is cell cycle blockade, cell lines were treated with different concentrations of compound for 16 h, then cell cycle was analyzed by flow cytometry. Results show that, depending on dose, chalcone blocks G2/M phase on A549 and H460 cells. Subsequently, role of LA in inducing apoptosis was evaluated using an annexin/propidium iodide colorimetric method. Results show that there is an accumulation of apoptotic cells in LA-treated group, which is dependent on concentration used. In addition, levels of proteins associated with apoptosis were examined by Western blot method. Levels of cleaved PARP and cleaved caspase 3 are elevated, and pre-caspase 3 antiapoptotic proteins, PARP, Bcl-xL, and Bcl-2 are decreased 20h after chalcone treatment. These results suggest that LA-induced apoptosis is associated with PARP/Bcl-2 pathway [177]. Molecular models showed that LA was docked in ATP binding pockets of EGFR, including exon 19 deletion mutation, L858R single-site mutation, L858R/T790M double mutation, and wild-type. In L858R/790M mutations, LA has ability to interact with Lys745 through cation–Π interaction. Hydrogen bonds are formed between WT EGFR and licochalcone on Met793, Lys745, and Asp 855.Exon 19 deletion has the ability to change the shape of the pocket in which interaction with licochalcone is considered to occur, forming hydrogen bonds with Met793, Thr790, and Glu762. Data obtained by in silico analyses of LA are an important point for identification of new, selective EGFR inhibitors on different types of mutations [178].

#### 3.1.3. Licochalcone B

Anticancer effects of licochalcone B (LB, Appendix A, compound 3) were demonstrated by analysis on different cell lines, including human cells from bladder cancer T24 and EJ, oral cancer cells HN22 and HSC4, MCF-7, A375 (a human melanoma cell line), and A431 (squamous cell carcinoma). Studies have shown that LB influences growth of cancer cells, inhibits formation of metastases, blocks cell cycle, and induces apoptosis [161]. Kang et al. investigated molecular mechanism by which LB induces apoptosis in human melanoma and cells in squamous cell carcinomas. Licochalcone has been shown to induce apoptosis of A375 and A431 cells by both intrinsic and extrinsic pathways. In case of testing antiproliferative effect of chalcone with trypan blue staining, it was observed that LB induces a significant decrease in cell viability, this decrease being correlated with concentration. After addition of LB, significant changes in cell characteristics were observed, including cell contraction, rupture of cell membranes, and an increase in percentage of fragmented nucleus cells. Increased percentages of pre-G1-phase cells and apoptotic cells were also noted [38]. Another study showed that LB significantly blocks cell cycle in the G2/M phase in case of HepG2-type cancer cell lines, and in case of bladder and breast tumor cells, compound blocks S phase [179]. Song et al. highlighted suppressive effect of LB on growth of JAK2-type esophageal carcinoma squamous cells. Docking studies were performed with Autodock Vina software, which was used to predict binding mode. Structure of the JAK2 receptor with inhibitory potential is available in Protein Data Bank (PDB entry 2B7A, residues 840-1.132). JAK2 plays an important role, being an intracellular mediator of cytokine signaling, and is a tyrosine kinase protein of JAK family. ATP binds strongly to magnesium ions in catalytic domain of tyrosine kinases. In the docking parameter, size of investigated space includes ATP binding site characterized by residues 855-863 and 822, where ATP was calculated with various potential JAK2 inhibitors. In predictions made, LB ligand was made with Marvin Sketch software. After docking, the best three possible binding variants were collected, which have similar affinities. Conclusions of predictions were favorable regarding interaction of LB with ATP binding pocket at JAK2 [180,181].

#### 3.1.4. Licochalcone C

Licochalcone C (LC, Appendix A, compound 4) is known to decrease inflammatory response on monocyte cell lines. This is due to a reduction in iNOS expression and restoration of antioxidant network activity of superoxide dismutase, catalase, and glutathione peroxidase. Kwak et al. conducted a study to describe relationship between ROS, c-Jun NH2 terminal kinase (JNK), and p38mitogen-activated protein kinase (MAPK) and established impact of LC in inducing apoptosis on KYSE 30 and KYSE450 esophageal cancer cell lines. Previous studies have determined IC50 values for LC treatment (45 μg/mL) after 24 h to inhibit proliferation of A549, MCF-7, and T24 cell lines. Inhibitions of 40, 47, and 68% were obtained for three cell lines. Kwak et al. obtained a dose- and time-dependent in vitro inhibition of esophageal cancer cell proliferation. From five cell types analyzed, KYSE30 and KYSE450, which have a common genetic support, had a similar response to LC treatment. In an analysis of anchorage-independent growth in soft agar, results indicate a significant reduction in ability of KYSE30 and KYSE450 cells to form colonies. Depending on concentration, chalcones induced apoptosis in both cell lines. Compound also induced an upward regulation of p24 and p27 (negative transition regulators in G1 and S phases of cell cycle) and regulated downstream cyclin D. LC also increased ROS generation in KYSE30 and KYSE450 cells. ROS activate mitogen-activated protein kinase (MAPK) pathway and induce cell apoptosis. In addition, compound increased level of JNK, c-Jun, and p38 phosphorylation and activated apoptotic pathways [182]. Similar to docking study by Song et al. for LB [180], Oh et al. highlighted binding interactions between LC and human JAK2 cells. Docking simulation was performed using Autodock Vina. To initiate docking study, structure of JAK2 receptor, which was solvated by an X-ray experiment, was obtained fromthe Protein Data Bank (PDB entry 2B7A). Structure of LC ligand was modeled by Marvin Sketch software and optimized by Chimera software. Catalytic site of JAK2 was correlated with a hinge region (residues 929-935), a DFGloop (residues 994-996), and a Ploop (residues 858-865). Loop-shaped hinge region is essential for recognition of ATP and forms hydrogen bonds with substances. DFGloop contains three amino acids (aspartic acid, phenylalanine, and glycine) and is associated with binding of a metal required for catalytic phosphorylation. Ploop is useful for stabilizing and forming interactions with ligands. As can be seen, prediction of a possible binding was made in three functional places. Docking study showed that LC interacts with ATP binding site to JAK2 and indicated that JAK2 is a direct target of it. Chalcone also suppressed JAK2 autophosphorylation by binding to ATP pocket of p-JAK2 [183,184].

#### 3.1.5. Licochalcone D

Licochalcone D (LD, Appendix A, compound 5) is an active flavonoid isolated from *Glycyrrhiza inflata*. A study was performed to evaluate ability of LD to inhibit cell proliferation by two targets for lung cancer cells (EGFR and MET) using sensitive and gefitimib-resistant human cells. To understand direct binding of chalcone to EGFR and MET, gefitimib-sensitive cell lines (HCC827) and gefitimib-resistant cell lines (HCC827GR) were used. Results of evaluations show that flavonoid binds to two receptors, suppressing activity of EGFR and MET kinases as a competitive inhibitor of ATP. In EGFR complex, chalcone has two hydrogen bonds formed by Met793 as main point and lateral corner of Asp855 in DFG loop. 4-Hydroxy-3-(3-methylbut-2-enyl)phenyl group and 3,4-dihydroxy-2-methoxyphenyl group are fixed on the same plane and blocked between hydrophobic residues Leu718, Val726, and Ala743 of P loop and Leu 844. In the Met complex, the keto group of the chalcone forms a hydrogen bond with Met1160. Tyr1159 as the main point and Ile1084, Val1092, Ala1108, and Lys1110 of P loop were covered similarly with a cap. LD is also strongly supported by side hydrophobic chains of Met1160′s main point and Leu1140, Met1211, and Ala1221 of lower ATP pocket. EGFR binding position closely resembles MET binding position, forming hydrogen bonds and a hydrophobic interaction. Chalcone is located identically in binding region for the two receptors. Stabilization of complex can be increased by a hydrophobic interaction. Predicted results were compared with experimental data, showing that flavonoid competitively inhibits the two receptors [185].

#### 3.1.6. Xanthohumol

Prenyl chalcones, due to their structural diversity, have different biological properties, including anti-inflammatory, anticancer, and antimutagenic activities [186]. Studies have shown that natural chalcones with prenyl groups have the potential to interfere with p53. For example, treatment of A549 cells with prenyl chalcone xantohumol (XN, Appendix A, compound 6) induces apoptotic cell death and blocks cell cycle in G1 phase. These activities are due to upregulation of p53 and p21 from cell cycle and downregulation of cyclin D1. Apoptosis is induced by activation of caspase 3 [187].

XN ((3′-(3,3-dimethylallyl)-2′,4′,4-trihydroxy-6′methoxychalcone) is the most abundant prenylated flavonoid (0.1–1% by dry weight) of female hop inflorescences (*Humulus lupulus*) [180]. XN is also a constituent of beer, a major dietary source of prenylated flavonoids, where it is present in concentrations above 0.96mg/L. Due to its unique biological activities and its favorable impact on health, prenylchalcone has been widely studied recently [188]. Compound has therapeutic safety and various bioactivities, including anti-cancer, anti-diabetic, anti-inflammatory, antioxidant, and antibacterial properties. In recent years, an increased number of studies have demonstrated broad spectrum of anticancer activity of XN in lung cancer, hepatocellular carcinoma, breast cancer, leukemia, prostate cancer, pancreatic cancer, colon cancer, pancreatic cancer, and glioblastoma cancer. Exposure of cancer cells to XN inhibits their proliferation, migration, and invasion and modulates autophagy. Chalcone also has ability to induce apoptosis and block cell cycle [189,190,191,192,193]. In addition, chalcone induces apoptosis dependent on and independently of caspase activity and inhibits cancer cell invasion and angiogenesis [194]. Its anti-inflammatory, antioxidant, and anticancer properties are correlated with chemopreventive effect of compound [195]. Prenylchalcone is also metabolized to 8-prenylnaringenin, the most potent phytoestrogen known to date [196].

Akt (also called protein kinase B or PKB) is a specific serine/threonine protein kinase and an important point in cellular signaling pathways. Akt activity is altered in many types of cancers and involved in various biological processes, including cell proliferation, apoptosis, transcription, migration, and invasion.To confirm XN’s ability to bind to Akt, an in silico docking study was performed using Schrodinger Suite 2015 software. Water molecules were removed, and pH for hydrogen atoms considered was 7. An ATP binding site was generated for docking study. XN was prepared for docking in absence of parameters using LigPrep program. Afterwards, docking studies of XN with Akt1 and Akt2 were accompanied by absent parameters using method of additional precision with Glide program in order to obtain the best structural representations. Results of docking study show that XN forms hydrogen bonds with Ala230, Glu228, Glu234, and Lys158 of Akt1 and with Glu236, Thr213, and Lys181 of Akt2. Xenograft (PDX) models were identified for translating basic research studies into clinical applications.To an increased extent, biological and genetic characteristics of donor patients are considered to be preserved by PDX models, which is the major advantage over cell-line-based models. PDX models were used to analyze biomarkers and to predict response to XN therapy in clinical trials. Chemopreventive effects of prenylchalcone were compared based on Akt levels. Results show that tumor models expressing a high level of Akt have a significant decrease in tumor volume and weight when treated with XN [197]. Guo et al. studied in vitro and in vivo effect of XN in gastric cancer, showing that prenylchalcone induces apoptosis by activating caspases, regulating Bcl-2, and influencing PI3K/Akt/mTor kinase. XN inhibits viability of gastric cancer cells in a concentration-dependent manner. On cell lines, flavonoid exerts the best effect on viability of SGC-7901 cells and does not influence this parameter on GES-1 cells at 6, 8, and 10 μg/mL of chalcone. From flow cytometric analysis, it was observed that prenylchalone significantly increases number of apoptotic cells in gastric cancer. Effect of XN on pro- and anti-apoptotic proteins was highlighted by Western blot analysis. Bcl-2 and Bcl-XL protein levels decreased after flavonoid administration, this decrease being correlated with administered concentration. XN also increased Bax and Bid protein levels, with the best activity being observed for 10 μΜ/mL of chalcone. In addition, levels of cleaved caspase 3 and cleaved PARP protein increased significantly in presence of chalcone. For these reasons, it can be stated that flavonoids favorably and significantly influence levels of pro- and anti-apoptotic proteins. A total of 10 μΜ/mL of XN induces a significant apoptosis of SGC-7901 cells. A total of 8 and 6 μΜ/mL of chalcone induce apoptosis of 34 ± 3% cells and 23 ± 2% cells, respectively. Additionally, flavonoid significantly modifies phosphorylation of PI3K, Akt, and mTOR, increases level of p-PTEM, and decreases level of p-Akt (Thr308), p-Akt (Ser473), and m-Tor (Ser2448). Resulting data indicate that prenylchalcone does not significantly influence Akt, PTEN, GSK-3β, and mTOR levels. Determinations in SGC7901 xenograft mice showed that XN treatment decreased tumor volume in a relatively concentration-dependent manner. To confirm suppression of PI3K/Akt signaling in vivo, phosphorylated Akt and mTOR expression on xenographic tumors was evaluated. Pathological examination of hematoxylin and eosin sections revealed significant morphological abnormalities. However, XN reduces concentration-dependent phosphorylated Akt and mTOR levels. Prenylchalcone treatment significantly decreased cell proliferation and increased tumor cell apoptosis compared with control cells [198].

XN, at concentrations higher than 10 μmol/L, inhibits proliferation of pancreatic cancer cells in vitro. At concentrations lower than 5 μmol/L, chalcone inhibits NF-ĸB-dependent angiogenic activity in pancreatic cancer cells. At this concentration, no cytotoxicity was observed on pancreatic cells by WST-1 method. However, conclusion of study was that XN influences pancreatic-cancer-induced angiogenesis by downregulating production of VEGF and IL-8 (an interleukin), which is specific and mediated by NF-ĸB inactivation [194].To evaluate XN’s anticancer activity, HepG2 cell lines were subjected to MTT analysis to determine cell proliferation. Prenylchalcone reduced cell proliferation depending on concentration and time. Zhao et al. observed that exposing cell lines to 200 μM of XN for one day is less effective compared with treating them with 100–200 μM of chalcone for 2–3 days. At 50 μΜ of chalcone, a significant inhibition of HepG2 cell proliferation was observed after 3 days. In same study, it was shown that prenylchalcone causes a significant increase in caspase 3 activity. Additionally, by Western blot analysis, it was shown that 100–150 μM of XN significantly inhibited expression of NF-ĸB protein on cell lines. By this analysis, it was also observed that prenylchalcone has capacity to increase expression of p53 protein, and 20 μM of XN determined an intensification of Bax signaling, this being correlated with time [199]. Studies of safety profile of XN show that 1000 mg/kg of compound does not alter functioning of vital organs and homeostasis in mice. Prenylchalcone has ability to increase IL-2 production in T cells, which demonstrates its ability to promote a Th1-mediated immune response. XN also inhibits IL-12, which indirectly produces differentiation of Th1 cells in the immune system by activating transcription molecules. Cytotoxic T lymphocytes are a type of cellular effector crucial to cellular immunity and play an important role in process of antitumor immunology. CD8^+^T cytotoxic lymphocytes exist as CTL-P, an inactive cell precursor in vivo. This precursor is activated by antigen in presence of Th1 cytokines, and then develops into mature cytotoxic T lymphocytes. A significant increase in CD8^+^/CD25^+^ was demonstrated, followed by transition of Th2 to Th1 in tumor microclimate. CD8^+^/CD25^+^ ratio of T cells is greatly increased when cytotoxic T lymphocytes are activated by CoCl2 on 4T1 cell lines. Function of Th1 and Th2 cells is dependent on secretion of various cytokines. To investigate effects of XN on Th1 and Th2 cytokines, Zhang et al. determined serum levels of Th1 and Th2-associated cytokines using ELISA kits. Prenyl derivative has been shown to significantly increase Th1 cytokine expression (including IL-2 and IFN-γ) and decrease Th2 cytokine levels (including IL-4 and IL-10). This conclusion is explained by fact that Th1 and Th2 are mutual inhibitors. In addition, Th1/Th2 ratio was determined by flow cytometry, showing that it is significantly increased by XN. Similar studies have reported this finding for various tumors. Patients with advanced squamous cell carcinomas of neck and head have low levels of Th1 cytokines compared with patients who are less severe and have high levels of Th2 cytokines. Combination therapy produces the transition of Th2 to Th1 cytokines in the tumor environment. Results of studies indicate a disorder of Th1/Th2 cytokine ratio, this change being observed for several types of tumors, most frequently in terminal stages of cancer. To confirm potential mechanism of XN on Th1/Th2 cytokine ratio, expression of key factors in pathway of Th1 and Th2 differentiation was determined. Physiologically, Th0 cells are proportionally differentiated into Th1 and Th2 cells. In addition, activation of transcriptional molecules 4 and 6 plays a vital role in differentiating Th0 to Th1 and Th2 cells. T-bet and GATA-3 also play two pivotal roles. CpG-ODN (cytosine–phosphorothionate–guanine contaning an oligodeoxynucleotide), a potent Th1 adjuvant, decreases GATA-3 expression and activation of transcriptional molecule 6 by activating T-bet and transcriptional molecules 1 and 4 in lung cancer models. XN increases T-bet expression and decreases GATA-3 expression. Activation of transcriptional molecule 4 is increased in the presence of XN, but it does not influence the activation of transcriptional molecule 6. For this reason, it can be stated that activation of transcriptional molecule 4 plays a positive role in regulating the Th1/Th2 cytokine ratio by XN [200].

Notch signaling pathway plays a significant role in breast cancer, which is a therapeutic target for its treatment. It is involved in initiation and progression of breast cancer, aberrant activity of this pathway being associated with this pathology. Inhibition of Notch signaling pathway by gamma secretase inhibitors and by anti-delta-like monoclonal antibody 4 is favorable for treatment of acute lymphoblastic leukemias and solid tumors. Mechanisms of these agents include cell cycle blockage or apoptosis and disruption of angiogenesis. Sun et al. investigated therapeutic potential of XN on breast cancer cell lines, highlighting its ability to inhibit cell proliferation, block cell cycle, and induce apoptosis in vitro. A reduction in tumor growth in vivo was also determined. In addition, possibility of prenylchalcone to inhibit growth of human breast cancer cells by the Notch signaling pathway has been investigated. To determine whether XN targets the Notch signaling pathway, a functionalized Notch 1 method was used, using a gamma secretase inhibitor (DAPT) as a control. Aim of study was to evaluate possibility that prenylchalcone reduces binding activity of Notch1 to CBF1 transgene. XN has been shown to inhibit proliferation and induce apoptosis by inhibiting Notch 1 pathway. Additionally, by MTT method and light microscopy, it was shown that prenylchalcone inhibits cell proliferation on breast cancer cell lines. Previous studies have shown that Notch pathway inhibitors are also inhibitors of EGFR expression, another incriminating element in breast cancer. In addition, XN acts on proteins associated with tumor metastases and inhibits cell migration by increasing the expression of these proteins. A study highlighted the blockade of cell cycle in G0/G1 phase and induction of apoptosis for MCF-7 and MDA-MB-231 cells by XN [201].

#### 3.1.7. Panduretin A

Anticancer activity of panduretin A (PA, Appendix A, compound 7), a cyclohexanylchalcone isolated from *Boesenbergia pandulata*, has been studied. Plant contains prenyl chalcones and other flavonoids as major bioactive molecules, which are described in literature to have preferential cytotoxic properties on human pancreatic cell line PANC-1. [202,203] PA is active in melanoma, colon adenocarcinoma, and prostate cancer. Proteomic analyses show that PA has cytotoxicity on melanoma cells that is dependent on denaturation of mitochondrial oxidative phosphorylation process, with activity of secretory pathway and apoptosis induced by oxidative processes. In this regard, it has been shown that oxidative stress can be result of stimulating autophagy as a secondary response to elevated ROS [204]. Literature indicates that a concentration of 9 μg/mL of PA completely inhibits the growth of MCF-7 cells and HT-29 cells (a human colon cancer cell line) [205].The chalcone has anticancer properties on various cell types, including melanoma, colon adenocarcinoma, and prostate cancer [204].Liu el al. highlighted the cytotoxic effect of the chalcone on MCF-7, T47D (human breast cancer), and MCF-10A (non-tumor breast cells) cell lines. The IC50 values of PA on MCF-7 cells were 15 μM at 24 h and 11.5 μM at 48 h. In the case of T47D cells, the IC50 was 17.5 μM at 24 h and 14.5 μM at 48 h. PA does not influence the proliferation of MCF-10A cells.To identify mechanisms by which the chalcone induces cell cycle blockade in MCF-7 cells in the GO/G1 phase, Western blot analysis was used, which aimed to evaluate the modulation of regulatory proteins in the cell cycle. Results show that PA treatment induces a decrease in cyclin D1 and CDK4 expression and increases p21Cip1 and p27 expression, thus explaining the blockade in the G0/G1 phase. Isolated PA from *Kaempferia pandulata* induces cell cycle blockade in androgen-independent PC-3 (prostate adenocarcinoma) cells and in human DU145 (a human prostate cancer cell line) cells.Internucleosomal DNA fragmentation is a marker of apoptosis. Because low-molecular-weight DNA fragments are extracted by staining cells in aqueous solutions, apoptotic cells can be identified by frequency histograms of DNA content in the form of cells with fractionated DNA content. A sub-G1 phase MCF-7 cell population was analyzed. The G1 phase content of cells was 1.17 ± 0.11 and in cells treated with PA (10, 15, and 20 μM) it was 1.84 ± 0.18, 2.62 ± 0.21, and 4.52 ± 0.28, respectively.The increase in the chalcone treatment was due to intensification of DNA fragmentation on MCF-7 lines, a fact confirmed by the sub-G1 phase cell population [206].

Among the key proteins of cancer cell invasion and metastasis induction are matrix metalloproteinases. They degrade components of the extracellular matrix and facilitate the invasion and migration of cells.In addition, overexpression of metalloproteinases may induce epithelial–mesenchymal transition. PA suppresses the secretion and activation of metalloproteinase 2, causing inhibition of endothelial cell migration, invasion, and morphogenesis on human umbilical vein endothelial cell (HUVEC) cells. In addition, subtoxic doses of chalcones are sufficient to downregulate metalloproteinase 2 in lung cancer cells [207].

#### 3.1.8. Cardamonin

Cardamonin (CD, Appendix A, compound 8), a chalcane from *Campomanesia adamantium* (Myrtaceae), increases DNA fragmentation and decreases NF-ĸB activity in PC-3 cells. These results indicate the therapeutic potential of the chalcone in the treatment of prostate cancer [20]. CD is considered one of the most active antitumor compounds in which activation of Epstein–Barr virus is involved [208]. Anticancer effects of CD are correlated with induction of apoptosis, inhibition of cell proliferation and migration, and an influence on the cell cycle. The chalcone also has the ability to reduce the resistance of cancer cells to therapy. In combination with 5-fluorouracil or *cis*-platinum, increased antitumor activities are obtained. For example, CD has the ability to significantly inhibit the resistance to chemotherapy of colon cancer cells, induces apoptosis, activates caspases 3 and 9, facilitates Bax protein expression, significantly inhibits c-myc, and carries specific 50 and NF-ĸB [209].Hou et al. investigated the therapeutic potential and molecular mechanisms of CD on 5-fluorouracil-resistant gastric cancer cells. The sensitivity of BGC-823/5-fluorouracil to 5-fluorouracil was confirmed by increasing apoptosis and blocking the cell cycle in the presence of CD. The chalcone increases the sensitivity of cancer cells to 5-fluorouracil by suppressing the Wnt/β-catenin signaling pathway (which plays a significant role in tumorigenesis), and activated mutations in Wnt/β-catenin genes are associated with resistance to anticancer therapy. It inhibits the expression of P-glycoprotein, β-catenin, and TCF-4.Additionally, CD specifically blocks the formation of the β-catenin/TCF-4 complex, thus causing aberrant Wnt/β-catenin signaling [210]. Badroon et al. investigated the antiproliferative and apoptotic effects of CD on HepG2 cells. The inhibitory action of the chalcone on HepG2 cell proliferation was significant after 72 h, the cytotoxicity being similar to that of 5-fluorouracil. Values determined for other chemotherapeutic agents used as standards (e.g., sorafenib) were much lower.In addition, the cytotoxic effect of the compound is selective on tumor cells and does not negatively influence normal cells, which is an advantage of CD compared with 5-fluorouracil. Accumulation of CD in the G1 phase of the cell cycle was observed after 72 h and indicates inhibition of HepG2 cell growth by preventing cell division [211].

Comparative docking studies of CD and 5-fluorouracil and its interaction with BaxBH3 show that 5-fluorouracil has a higher binding energy than CD. The chalcone forms three hydrogen bonds (Phe30, Val50, and Gln52).Interaction between CD and Bcl is achieved by three hydrogen bonds (Asp15, Gln18, and Ser28), and in the case of 5-fluorouracil is achieved by four bonds. Additionally, in the case of this interaction, the binding energy of CD is lower than in the case of 5-fluorouracil. This can be attributed to aromatic residues in the structure of the chalcone, which are involved in Π bonds, which have the ability to stabilize the active pocket and cause a decrease in the binding energy. Results of in silico studies show that 5-fluorouracil has higher binding energies than caspase 3 compared with CD.CD shows two hydrogen bonds in the interaction with caspase 3 (Cys163 and Arg64). The chalcone also has Π–Π bonds with TYR204. The free binding energy of 5-fluorouracil is superior to that of CD, which is explained by the stabilization of the active pocket by two aromatic residues in the structure of the chalcone [212].

#### 3.1.9. Lonchocarpin

Lonchocarpin (Appendix A, compound 9) is a natural chalcone extracted from *Lonchocarpus sericeus*. Cytotoxic effects of this chalcone have been described on neuroblastoma and leukemia cell lines. It is known that 24 h after treatment with 50 μM lonchocarpin on SK-N-SH neuroblastoma lines, induction of AMPK phosphorylation takes place, which increases glucose absorption and inhibits protein synthesis. The chalcone also has the ability to decrease cell viability. On colorectal cancer cell lines HCT116, SW480, and DLD, lonchocarpin reduces cell viability by 20 μM. Studies show that lonchocarpin has the ability to inhibit H292 lung cancer cells in vitro by the caspase-3-induced cell death that precedes apoptosis. In addition, lonchocarpin has been observed to inhibit Wnt/β-catenin signaling in vivo in embryonic models of *Xenopus laevis*. The chalcone’s injection into a co-injected Wnt8-specific receptor model (SO1234) resulted in 82% suppression of Wnt/β-catenin signaling receptor gene activation [213].

In study by Chen et al., results of 3D-QSAR analysis indicate a hydrophobic C-4, C-5, C-11, C-1‘, and C-2′ interaction in lonchocarpin. This interaction increases the cytotoxic capacity of this compound, having a contribution of 23% in the model. Docking studies for lonchocarpin yielded the same results as the hydrophobic 3D-QSAR model, with the hydrophobic surface in the C-4, C-5, C-11, C-1‘, and C-2′ regions of loncocarpine interacting with the Bcl-2 complex. The hydrophobic loop of the Bcl-2 protein forms a complex with the BaxBH3 peptide, which can be interrupted by a synthetic novitoclax or lonchocarpin compounds.This shows that the hydrophobic loop of Bcl-2 family members is a target for loncocarpine-induced apoptosis on H292 cells and thus for activation of caspase 3 [214].

Other natural chalcones with anticancer properties are butein (Appendix A, compound 10), isoliquiritigenin (Appendix A, compound 11), flavokawain (Appendix A, compound 12), and isobavachalcone (Appendix A, compound 13) [155].

## 4. Synthetic Derivatives of Chalcones with Anticancer Properties

Anticancer activities of natural chalcones have led to increased interest in identifying new synthetic chalcones with anticancer properties. Objectives for the synthesis of new biologically active chalcones are the identification of compounds with superior physico-chemical and biological properties. To obtain chalcones with superior anticancer properties, three methods of modulation of natural chalcones were used:(1) modulation of the two aromatic residues (the aldehyde and the acetophenone) of chalcones;(2) replacement of aromatic residues with heteroaromatic residues; and (3) obtaining hybrids by conjugation with other molecules with antitumor properties. Different substituents on the two aromatic residues of chalcones, depending on their position, influence the anticancer capacity by interfering with different biological targets [158]. It is known that biological properties of chalcones are dependent on the presence and number of hydroxy and methoxy groups on the two aromatic subunits. For example, chalcones with three methoxy groups in the molecule on the 3, 4, and 5 positions of the acetophenone inhibit the transport activity of P-glycoprotein and prevent the onset of resistance to therapy [155,159].

### 4.1. XN Acyl Derivatives

Starting from the idea that esterification of flavonoids is a way to modify the hydrophobic character of compounds, a series of mono- and diacetylated derivatives of XN (compounds 14–20, Appendix A,) were synthesized by Żołnierczyk et al. The antiproliferative activities of XN and its derivatives were tested in vitro on HT-29 cell lines. Three compounds from the series (compounds 14–16) showed XN-like bioactivities and four tested compounds (compounds 17–20) had lower bioactivities. From the obtained series, no compound had higher activity than XN [215].

Another series of XN derivatives was obtained by cyclizing the prenyl group from its structure. Thus, a series of six cyclic derivatives of chalcones (Appendix A, compounds 21–26) were obtained by Popłoński et al. The antiproliferative activity of the obtained compounds was evaluated on three human cell lines (MCF-7, PC-3, and HT-27). The potency of the XN derivatives was evaluated by the SRB method. All compounds obtained showed moderate/increased bioactivity, the most vulnerable cell line being MCF-7. Compounds 21and 23((E)-1-(5-hydroxy-7-methoxy-2,2-dimethyl-2H-chromen-6-yl)-3-(4-hydroxyphenyl)prop-2-en-1-one and(E)-1-(5-hydroxy-7-methoxy-2,2-dimethylchroman-8-yl)-3-(4-hydroxy phenyl) prop-2-en-1-one) showed the best activity on PC-3 lines, their action being comparable to the activity of the standard (*cis*-platinum) [216].

### 4.2. Chalcone Derivatives Containing a Diaryl Ether Moiety

Wang et al. synthesized chalcone derivatives with a diaryl ether residue (Appendix A, compunds 27–42) in the molecule and evaluated their antiproliferative activity on three cell lines (MCF-7, HepG2, and HCT116). Results show that most of the compounds have a moderate/good activity, on the three cell lines, with an IC50 between 3.44 ± 0.19 and 8.89 ± 0.42 μM. From the obtained series, the compound substituted with 4-methoxy on the aldehyde (Appendix A, compound 28) is the most active compound (IC50 = 3.44 ± 0.19, 4.64 ± 0.23, and 6.31 ± 0.27 μM on MCF-7, HepG2, and HCT116, respectively). Replacement of the 4-methoxy group with 4-dialkylamino (Appendix A,, compound 29) resulted in a significant decrease in activity.This compound is a potential inhibitor of tubulin polymerization, with a colchicine-like mechanism. Additionally, 4-methoxychalcone (compound 28) has antiproliferative properties on MCF-7 cells by increasing the percentage of cells in the G2/M phase. In addition, the chalcone induces apoptosis of MCF-7 cells, as determined by the Annexin V-FITC/PI method. Docking studies indicate binding energies of −8.0 kcal/mol for binding the tubulin compound 28, in the pocket of which it adopts a Y-shaped conformation. The 4-methoxy and trimethoxyphenyl groups of the compounds form strong hydrophobic bonds with residues Ala180, Cys241, Leu248, Ala250, Leu255, Ala316, Val318, and Ala354. Additionally, the phenyl group of the compounds forms a cation–Π interaction with the Lys254 residue. In addition, the compound forms two hydrogen bonds with residues Asn101 and Ser178. These interactions facilitate the anchoring of compound 28 to the tubulin binding site [150,217].

### 4.3. Chalcone Derivatives Containing a Sulfonamide Moiety

α,β-Unsaturated derivatives of sulfonamide (Appendix A, compounds 43–54) were obtained and physico-chemically characterized by Castano et al. From series of compounds, compounds 43, 44, 45, and 50 hada cytotoxic effect at 10μM. All hybrid molecules were active on the HTC-116 cell line (−78.33–44.62%) and U251 (a glioblastoma cell line, −4.20–35.40%).Compounds 44 and 50 were the most active on most cell lines (IC50 = 0.57–12.4 μM for compound 44 and 1.56–40.1 μM for compound 37). Chalcone 44 had the best activities on K562 leukemic cell lines (IC50 = 0.57 μM). The compound also had a good ability to inhibit HCT-116 lines (IC50 = 1.36 μM), LOX IMVI melanone lines (IC50 = 1.28 μM), and MCF-7 (IC50 = 1.30 μM) [218].

### 4.4. Bis-Chalcone Derivatives

Compounds that have two subunits of chalcones in the molecule are called bis-chalcones. Some bis-chalcones are cytotoxic agents on various human cell lines (A549, DU145, KB (a keratin-forming tumor cell line), HeLa, and KB-VN). Bis-chalcones with a biphenyl residue in the molecule are active on MCF-7, MDA-MB 231, HeLa, and HEK-293 (human embryonic kidney) cell lines. Starting from these premises, a series of eight bis-chalcones (Appendix A, compounds 55–62) were synthesized, whose anticancer activity was evaluated on the MCF-7 and Caco2 cell lines by the MTT method. All compounds from the series had superior *cis*-platinum activity on the tested cell lines.The bis-chalcone substituted with two fluoro groups in positions 2 and 5 (compound 61) had the best IC50 values on the MCF-7 cell lines (1.9 μM), indicating an approximately three-fold better activity than the other compounds from the series. Morphological changes determined on MCF-7 cells at 24 h by the bis-chalcone demonstrate a significant decrease in the level of cell confluence compared with other compounds. For the Caco2 cell lines, the results were similar to those for MCF-7. Additionally, compounds 61 and 62 had the highest toxicity on the cell lines, and compounds 58 and 59 had the lowest activity [140].

### 4.5. Chalcones with Nitrogen in the Molecule

Aminochalcones are known to have strong cytotoxic effects. For example, 2-aminochalcones with a methylenedioxy residue in the molecule show very good activity on human nasopharyngeal squamous cell carcinoma (KB-VIN) cell lines. In addition, another study indicates that unsubstituted 2-aminochalcones on the aldehyde have proapoptotic effects on 20 apoptotic markers [219].

Starting from the fact that different substituted 2-aminochalcones show cytotoxic activity on different cell lines, such as KB (nasopharyngeal squamous cell carcinoma), MCF-7, A-549, and 1A9 (ovarian cancer), and are inducers of apoptosis on HT-29 cells, a series of aminchalcone derivatives were obtained (Appendix A, compounds 63–80). The anticancer activity of the obtained compounds was evaluated on four cell lines (HT-29, LS180 (an intestinal human colon adenocarcinoma cell line), LoVo (a colon cancer cell line), and LoVo/Dx by the SRB method. The standards used were *cis*-platine and doxorubicin. Among compounds obtained, the best inhibitory capacity was exhibited by compound with an unsubstituted aldehyde (compound 63). Activity of compound on HT-29 cell lines was IC50 = 1.43 μg/mL, being 12 times higher than the activity of *cis*-platinum (IC50 = 16.73 μg/mL) and 4 times lower than the activity of doxorubicin (IC50 = 0.33 μg/mL). Fromthe 4-aminochalcones (compounds 75–80), the unsubstituted compound on the aldehyde (compound75) had the best activity. Similarly, the activity of 3-aminochalcones (compounds 69–74)varied onthe tested cell lines (IC50 = 1.60–2.13 μg/mL). The potency of these compounds was superior to that of *cis*-platinum. In the case of the aminocarboxylic derivatives (compounds 65, 71, and 77), the position of the amino group had a significant impact on the IC50 value. The activity varied in the following order: 2-amino (compound65) >3amino (compound 71) >4-amino (compound 77). It was also observed that the incorporation of a nitro group at position 4 of the aldehyde (compounds 66, 72, and 78) caused a decrease inactivity [220].

Series of aminochalcones and nitrochalcones were obtained in order to evaluate their cytotoxicity. Activity was determined by the MTT method on melanoma cell lines. Compared with nitrochalcones, aminochalcones (Appendix A, compounds 81–91) have the advantage of increased solubility in biological media. It was determined that substitution of chalcones with an amino group is favorable, the activity of these compounds being superior to that of nitrochalcones. The IC50 values showed that the presence of an amino group on the aldehyde caused an increase in cytotoxicity and amino group compounds on the acetophenone have a weaker activity. For example, compound 87 (in which the amino group is on the aldehyde residue) has a higher cytotoxicity than compound 86 (in which the amino group is on the acetophenone residue). Additionally, the number of methoxy groups on the acetophenone determines the inhibitory potency of these compounds. Data obtained indicate that aminochalcones substituted with two or three methoxy groups are more active. In the case of chalcones substituted with an amino group on position 3 of the aldehyde (compounds 87 and 90), the cytotoxicity is higher compared with amino-substituted compounds on position 4 (compounds 88 and 89). From the chalcones obtained, compound 87 (with an amino group on position 3 of the acetophenone and with four methoxy groups) had the best activity [221].

Wang et al. obtained a series of aminochalcones (compounds 92–103, Appendix A,) that were evaluated for anticancer activity on cell lines (HTC116 and HepG2) by the MTT method. All compounds were found to have a good/moderate cytotoxic capacity. The unsubstituted nitrogen compound (compound 92) had the best activity (IC50 = 0.28 ± 0.06 for HCT116 and 0.19 ± 0.04 for HepG2). Substitution of the amine with alkyl groups (compounds 93, 94,96, and 98) caused a significant decrease in antiproliferative activity. A marked decrease in activity was observed on the aminochalcone with two 4-(tert-butyl)benzyl residues (compound 99). Results obtained from the in vitro evaluation of the tubulin inhibition capacity for compound 92 indicate that its molecular target is tubulin, the IC50 value for aminocalcone being 7.1 μM and for colchicine being 9.0 μM. It was also observed that the aminochalcone (compound 92) had the ability to increase the proportion of cells in the G2/M phase and to block the cell cycle. Docking studies for compound 92 show that it binds to the binding site of colchicine in tubulin. The aminochalcone adopts an “L-shaped” conformation in the tubulin pocket. The 4-methoxynaphthyl group of the aminochalcone is located in the hydrophobic pocket, being surrounded by residues Cys241, Leu248, Ala250, Leu255, Ile318, and Ala354, with which it forms a strong hydrophobic bond [222].

Modification of the amino group in the structure of aminochalcones determines every time an increase in the anticancer activity of compounds [223]. Starting from this premise, we performed a literature study on the antitumor activity of some heterocyclic chalcones with a nitrogen in the molecule (azoles).

#### 4.5.1. Azoles

Azoles (imidazole, oxazole, pyrazole, tetrazole, thiazole, 1,2,3-triazole, and 1,2,4-triazole, Figure 8) constitute the most important class of nitrogen heterocycles. Azoles are important pharmacophores for the identification of new anticancer agents. Some azole derivatives (cefatrizin, carboxyamidotriazole, and AZD8835) are used clinically or are in clinical trials for the treatment of various cancers. Hybridization of chalcones with azoles is considered to be an important way to identify new anticancer agents [162].

#### 4.5.2. Imidazole

Imidazole (Figure 8), a five-atom heterocycle, has an increased polarity due to the presence of two nitrogen atoms. The system has an amphoteric character (it can have basic or acidic properties). Imidazole is known to be present in many biologically active compounds with anticancer properties [224,225]. Different substituted 2-benzimidazole derivatives are active on cell lines of breast adenocarcinoma, human hepatocellular carcinoma, and human colon carcinoma [226].

### 4.6. Imidazole Chalcone Derivatives

Oskuei et al. obtained imidazolechalcones (Appendix A, compounds 104–121) to evaluate their ability to inhibit tubulin. The antiproliferative activity of the compounds was evaluated on four different cancer cell lines (A549, MCF-7, MCF-7/MX (a mitoxatrone-resistant human breast cancer cell line), and HepG2). Many compounds of the obtained series showed medium/high antiproliferative activities at micromolar concentrations. In general, imidazole chalcones had a higher cytotoxicity on A549 cell lines compared with the other cell types analyzed. The compound substituted with three methoxy groups on the acetophenone (Appendix A, compound 121) had the best activity, which can be explained by the presence of a trimethoxyphenyl subunit as an important pharmacophore for potent tubulin inhibitors (e.g., combresatin A4, Figure 7). The increased cytotoxicity of theimidazolechalcone with a trimethoxyphenyl residue (compound 121) was due to its interaction with tubulin. Compounds in which the phenyl residue on the acetophenone was replaced by a naphthyl residue (compound 108, compound 117) hada good potency. This can be explained by the ability of these chalcones to penetrate the cell membrane due to the increased lipophilia. These compounds have favorable interactions with active sites of tubulin. Application of the tubulin polymerization method showed that the obtained imidazole chalcones inhibited, in a concentration-dependent manner, tubulin polymerization in a manner similar to combrestatin A4. In addition, the cytotoxicity of the most active compounds in the series was correlated with blockade of the cell cycle in the G2/M phase and induction of cellular apoptosis. Docking studies showed that an imidazolechalcone with three methoxy groups (compound 121) on the acetophenone had the best ability to bind to the colchicine binding site of tubulin. The compound has two interactions through hydrogen bonds with catalytically active residues (Ser178α and Ala316β) and a cation–Π interaction with Asn258β. Other hydrophobic interactions were observed between the compound and residues Glu183α, Thr224α, Lys254β, Asn101α, Val351β, Lys352β, and Leu248β. Hydrophobic interactions and hydrogen bonds formed between the compound and tubulin were found to be responsible for its inhibitory effect [227].

#### 4.6.1. Pyrazole

Pyrazole (Figure 8) is an important component of five-membered heterocycles in molecules. Two nitrogen atoms are in adjacent positions. Of these, one is basic and one is neutral. Numerous methods have been identified for obtaining pyrazole derivatives, which are important elements of medicinal chemistry. Studies have shown that some pyrazole derivatives have anticancer properties. For example, pyrazole is a pharmacophore for anticancer compounds such as Ruxolitinib (blood cancer), Axitinib (kidney cancer), and Crizotinib (lung cancer) [228,229,230,231].

#### 4.6.2. Pyrazole Chalcone Derivatives

A series of nine chalcones with a pyrazole in the molecule (Appendix A, compounds 122–130) were synthesized in order to evaluate their anticancer potential. Cytotoxicity was evaluated in vitro on A549 cell lines using the MTT method. The compound substituted with a trimethoxyphenyl residue on the acetophenone (compound 124) was the most active, its anticancer potential being present at micromolecular concentrations. Results obtained are in accordance with data from the literature indicating pharmacological activities of pyrazoles with a trimethoxyphenyl residue in the molecule (anticancer, antiproliferative, and antitubulin properties). For the obtained compounds, docking studies estimated binding interactions between Lys347, Lys356, and Glu354. Results indicate the presence of strong binding interactions between the methoxy group of the compound with a trimethoxyphenyl residue on the acetophenone and the hydrogen atom of Lys356, between the carbonyl oxygen with the hydrogen atom of Lys356 and LYS347, and between the hydrogen from benzopyrone with the atom of Lys4747 [232]. Hawash et al. obtained hybrid chalcone molecules with 1,3,4-trisubstituted pyrazoles (Appendix A, compounds 131–172) with a heterocycle. Bioactivities of the derivatives were analyzed for HCT116, hepatocellular (Hub7), and MCF-7 cell lines. In general, compounds that had a thienyl subunit in the 3rd position of the pyrazole (compounds 131–138) had a very good antiproliferative activity. Compounds with methoxy groups in positions 3 and 4 or 2 and 5 of the phenyl on the chalcone (compounds 135, 136, 143, 144, 160, and 170) had IC50 values of 0.4–3.4 μM on Hub7, MCF-7, and HCT116 cells. Replacement of the thienyl residue with benzo [d] [1,3] dioxo-5-yl (compounds 139–150) resulted in a significant decrease in cytotoxicity [233].

#### 4.6.3. Tetrazole

Tetrazole, an unsaturated double heterocycle with five atoms, contains four nitrogen atoms and one carbon atom. Biologically active substances with tetrazole in the molecule have an increased bioavailability, and replacement of the carboxylic acid with tetrazole causes an increase in bioavailability and a reduction in adverse effects. The tetrazole derivative letrozole is used clinically for the treatment of tamoxyfen-refractory breast cancers [234].

##### Tetrazole Chalcone Derivatives

Monaem et al. obtained a series of tetrazole chalcones (Appendix A, compounds 173–179). For the obtained compounds, cytotoxicity was evaluated by the MTT method on HCT116, PC3, and MCF-7 cell lines and on *Vero B* (African green monkey kidneys). Results were compared with *cis*-platin and 5-fluorouracil. Many of the compounds obtained had an activity higher than that of the standards on HCT-116 and PC-3 cell lines. Cyclization of chalcones to the corresponding pyrazolines resulted in a decrease in activity [235].

#### 4.6.4. Thiazole

Thiazole, a heterocycle derived from thiosemicarbazide, is present in compounds with antiparasitic, antifungal, and antiproliferative properties. Compounds with 1,3-triazole substituted in positions 2 and 4 are pharmacophores for tumor agents with significant activity. Thiazole derivatives have antiproliferative properties that correlate with inhibition of metalloproteases, some kinases, and Bcl2 family proteins [236]. Two heteroatoms (nitrogen and sulfur) have electron pairs that have the ability to form hydrogen bonds with amino acid residues of receptor proteins. These interactions are responsible for the apoptotic action of thiazole compounds on cancer cells. The heterocycle is a pharmacophore for anticancer agents such as epothilones, ixabepilone, bleomycin, thiazofurine, dasatinib, and kud773 [237].

##### Thiazole Chalcone Derivatives

Farghali et al. obtained thiazole chalcones (Appendix A, compounds 173–178). The antiproliferative activity of the obtained compounds was determined on three cell lines (HepG2,A549, and MCF-7). Compound 178 (3-(4-Methoxyphenyl)-1-(5-methyl-2-(methylaminothiazol-4-yl)propen-2-en-1-one) had superior anticancer activity to doxorubicin and a wide range of activities. The chalcone has IC50 = 1.56, 1.39, and 1.97μM on HepG2, A549, and MCF-7 lines, respectively, the values being half of the doxorubicin values (IC50 = 3.54, 3.19, and 4.39 μM, respectively). From the six thiazole chalcones, five compounds showed very good cytotoxicity on the cell lines tested, and the compound substituted with a 2,4-dichlorophenyl residue (compound 178) had a moderate level of activity. To evaluate the selectivity between tumor and normal cells, three chalcones with very good cytotoxic potential were tested on the non-cancerous lung cell line WI-38. Elevated IC50 values (93.44–137.36 μM) indicated selective cytotoxicity in malignant lung cells. The most safe chalcone was found to be the one substituted with 4-methoxyphenyl (compound 173). This chalcone inhibited HepG2, A549, and MCF-7 cells 88.04, 98.8, and 69.72 times more than WI38 cells. The derivative also significantly blockedthe cell cycle in the G2/M phase. The chalcone increased the DNA content in the G2/M phase by 2.6 times and decreased the amount of DNA in the G0/G1 and S phases compared with control cells. In addition, the compound caused a 14.3-fold increase in the percentage of pre-G1 cells compared with the control group, which indicated a possible role of the chalcone in apoptosis. The apoptotic capacity of the compound was assessed by the Annexin V-FITC method. The percentage of apoptotic cells increased significantly, indicating the ability of the compound to induce apoptosis. Docking studies of the three most active chalcones show that they bind to ATP at the CDK1 binding site, the binding energies being −6.373, −5.857, and 5.519. The compounds bind in the same way to amino acid Leu83, by forming two hydrogen bonds between the thiazole sulfur and between the 2-aminomethyl group and Leu83. In addition, two chalcones have the ability to form another hydrogen bond with the Glu81 residue of the target enzyme at the level of the sulfur on the thiazole [238].

Suma et al. obtained and physico-chemically and biologically characterized ten chalcones with a thiazolo-imidazopyridine residue in the molecule (Appendix A, compounds 179–188). The obtained compounds were tested on four cell lines (MCF-7, A549, DU-145 (a prostate carcinoma cell line), and MDA MB231 (a breast carcinoma cell line)). The method by which the anticancer activity was tested was the MTT method, and the standard used was etoposide. The most active compound from the series had three methoxy groups in positions 3, 4, and 5 of the acetophenone (compound 180). IC50 values of compound 180 for MCF-7, A549,DU-145, and MDA MB-231 were 0.18 ± 0.094 μM, 0.66 ± 0.071 μM, 1.03 ± 0.45 μM, and 0.065 ± 0.082 μM, respectively.

Compound with a single methoxy group on acetophenone (compound 182) had a much lower anticancer activity. From SAR studies, it was observed that presence of three methoxy groups (electron donors) determines a significant increase in bioactivity in case of thiazolo-imidazopyridine derivatives. Docking studies were performed on three potential targets: protein kinases CLK1 (5X81), EGFR (2J5F), and tubulin (1SAO). Scores obtained indicated a correlation between the activity of compounds and their action on CLK1 [239].

#### 4.6.5. Triazole

Triazole is a penta-atomic heterocyclic organic compound that contains three nitrogen atoms and two carbon atoms. It is present in two isomeric forms, 1,2,3-triazole and 1,2,4-triazole [240].The heterocycle is an important pharmacophore for molecules with anticancer, anti-HIV, anti-inflammatory, and antituberculosis properties. The compound 1,2,3-Triazole is a basic element of medical chemistry because it has the ability to form hydrogen bonds with important biological targets [241]. The compound 1,2,4-triazole also influences the lipophilia, polarity, and ability of molecules to form hydrogen bonds [242].

##### Triazole Chalcone Derivatives

Studies from literature show that 1,2,3-triazole–chalcone hybrid molecules have remarkable anticancer activities on SK-N-SH cell lines (IC50 = 1.52 μM) by inducing apoptosis [243]. Hybridization of 1,2,4-triazole ring with a chalcone also caused significant inhibition of cancer cell growth and induced apoptosis of A549 cells dependent on caspase 3 activity with an IC50 = 4.4 μM (compound 189) compared withcis-platin’s IC50 = 15.3 μM [244]. Gurrapu et al. synthesized 1,2,3-triazole chalcones (Appendix A, compounds 190–198) and their cytotoxicity was determined experimentally and in silico. The cancer cell lines on which cytotoxicity was determined were MCF-7, HeLa, and MDA MB231 and the method used was the MTT method. From the nine compounds tested, the triazole derivative with a chlorine in the meta position of the substituent attached to the triazole and two methoxy groups on the acetophenone (compound 196) showed the best activity on all lines tested (e.g., IC50 for MCF-7 = 1.27 μM and 0.02 μM at 24 and 48 h, respectively), the results obtained for this compound being comparable to those of *cis*-platin. A decrease in viable cells was observed by increasing the concentration. Results of application of cell viability method showed that triazole chalcones have a good oral bioavailability. Drug-likeness was determined by the number of rotating free bonds and rules of Lipinski, Veber, Eagan, and Mugge. All compounds from series had good pharmacokinetic profiles and satisfied criteria for drugs. Series comprised pharmacophores having a triazole nucleus bound to a residue -OCH2-. Compounds having electron donor groups, in particular molecules substituted with a chlorine in meta position of triazole ring and two methoxy groups in meta and para positions of acetophenone (compound 150), with a chlorine in the meta position of the substituent on the triazole and a hydroxy group in meta position of chalcone (compound 194), or a methyl in meta position of the substituent attached to triazole and with two methoxy residues (compound 193) were the most active cytotoxic agents from the series. Possible binding mode for obtained compounds was determined for EGFR kinase. Molecules had a range between −8.102 and −6.008 kcal/mol and values of binding energies between −83.05 and 43.696 kcal/mol. Compound with a chlorine in meta position of substituent attached to triazole and a hydroxy group in meta position of chalcone (compound 198) showed highest scores (−8.102 and −83.05 kcal/mol). This compound forms a hydrogen bond interaction with Asp800, a strong Π–Π interaction with Phe856 and Phe997, and a Π–cation interaction with Lys745. For all compounds from series, the phenyl attached to the triazole forms Π–Π interactions with Phe856. Phenolic hydroxy group forms interactions through hydrogen bonds with amino acid Asp800 [245].

## 5. Conclusions

Cancer is a disease triggered by many mechanisms and is a major public health problem.

Chalcones are precursors to all other flavonoids and to many other heterocyclic compounds. The advantages of these compounds are related to their numerous biological properties, their lack of adverse effects, the possibility of obtaining them easily, and the possibility of forming numerous biologically active compounds by modulating their basic structure. In addition, chalcones are a starting point for the identification of new anticancer compounds. Natural and synthetic chalcones have antitumor properties in vivo and in vitro and are also active in drug-resistant cancers.

An important mechanism of the antiproliferative activity of chalcones is inhibition of tubulin and interference of these compounds with the assembly of microtubules. Substitution of chalcones with three methoxy groups is favorable for their antitubulin activity, as these compounds have a structure that is similar to A4 combretastatin. Hybridization of chalcones with anticancer pharmacophores is favorable for their activity. For example, the introduction of an azole into a molecule of these compounds has led to a significant increase in their biological properties. This fact can be correlated with the binding of these compounds and with the favorable change in lipophilic parameters.

## Figures and Tables

**Figure 1 ijms-22-11306-f001:**
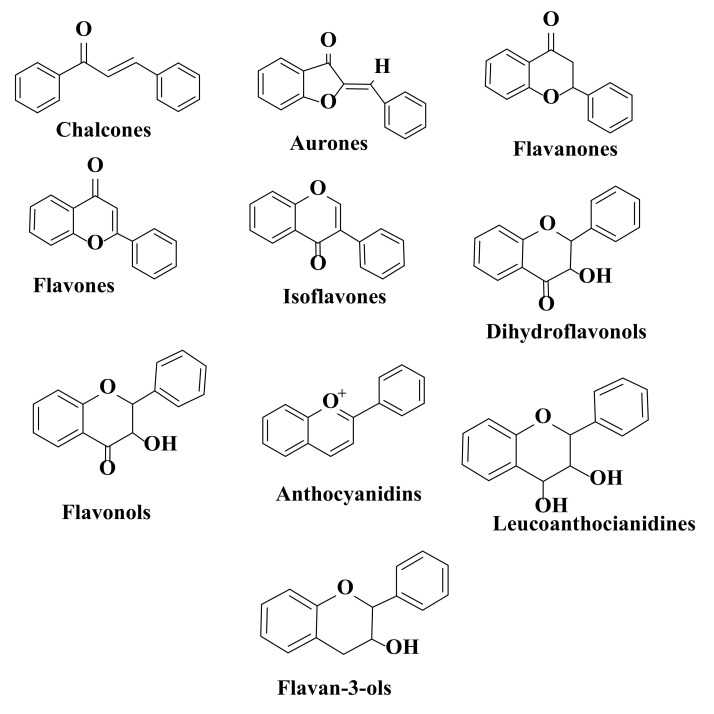
Basic structure of flavonoids.

**Figure 2 ijms-22-11306-f002:**
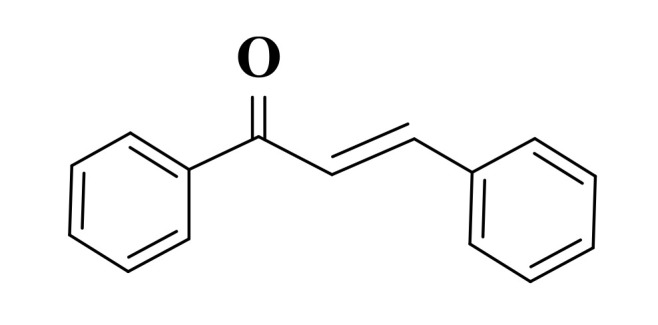
General structure of chalcones.

**Figure 3 ijms-22-11306-f003:**
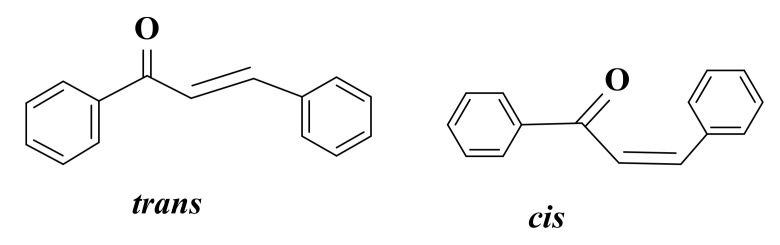
Cis and trans isomers of chalcones.

**Figure 4 ijms-22-11306-f004:**
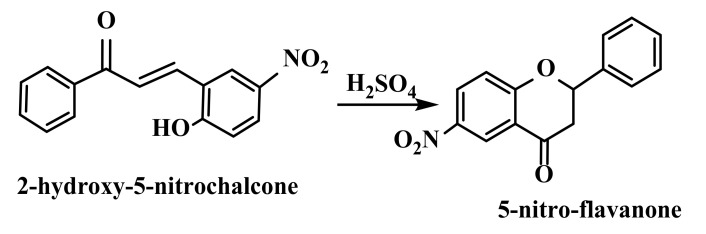
Cyclization of 2-hydroxy-chalcone to flavanones [116].

**Figure 5 ijms-22-11306-f005:**
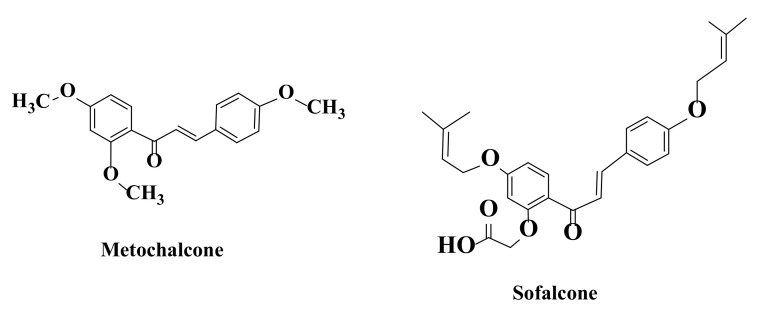
Structure of metochalcone and sofalcone.

**Figure 6 ijms-22-11306-f006:**
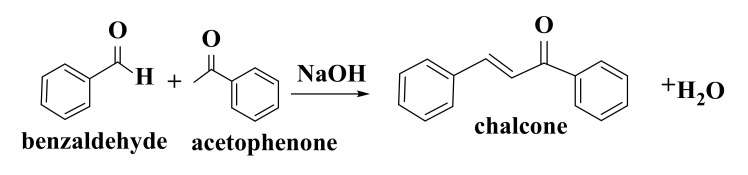
Claisen–Schmidt reaction.

**Figure 7 ijms-22-11306-f007:**
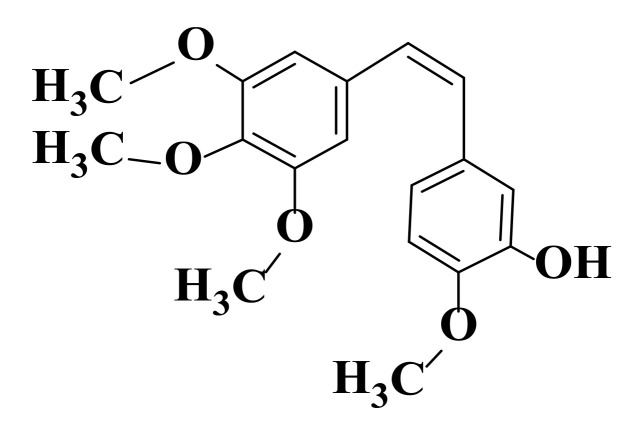
Combrestatin A-4.

**Figure 8 ijms-22-11306-f008:**
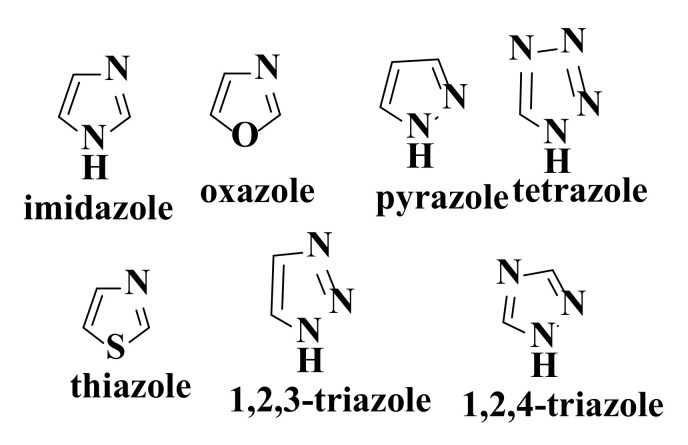
Azoles.

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
