# Peer review of "Anticancer Activity of Natural and Synthetic Chalcones"

_ijms, 2021, doi:10.3390/ijms222111306_

Round 1

Reviewer 1 Report

Review work written quite chaotically. I do not fully understand what the authors meant. They start with natural chalcones and then quickly move on to azolochalcones, devoting a whole chapter to them. There is no chapter or mention of how modifications of natural chalcones change anti-cancer activity, e.g. work 10.1007/s00044-017-1887-9 and others will be a valuable part of this review. There are also missing new group of aminochalcones with great anticancer, e.g. 10.3390/molecules24224129 and 10.1007/s00044-004-0122-7 and others. I also propose to make subsections for individual compounds, instead of putting all of them together. Please review the citation record - should there be any spaces between the dot and the citation brackets. Please unify all the formulas so that the bonds have the right angles and meet the requirements of IJMS MDPI. Supplementary Materials file missing!

Other remarks:

1) Figure 2: Incorrect substrate - missing 2'-OH group. Please correct

2) Figure 5: "Metochalcone" ??

3) Lines 215-219: why is one synthesis example given as general? This is not true, please correct it.

4) Lines 245-260: should this be in the introduction for sure?

5) Line 265: it should be "NF-κB"

6) Whole article: in vitro and in vivo should be italic.

7) Whole article: Latin names for plants should be italic.

8) Lines 271: should be "chalcones"

9) Whole article: there should be a space between a numeric value and a unit.

10) Whole article: it should be trans-chalcone.

11) Lines 682, 685: there should be π instead of pi

12) Lines 711-712: Surprising and unnecessary paragraph. Please either expand it or delete it. The more so that another review is quoted.

13) Lines 843-844: please check the spelling

14) Please rebuild the summary.

Author Response

Comment1 Review work written quite chaotically. I do not fully understand what the authors meant. They start with natural chalcones and then quickly move on to azolochalcones, devoting a whole chapter to them.

Response1: The review is structured accordingly to the mdpi  accepted review template.

Comment2 There is no chapter or mention of how modifications of natural chalcones change anti-cancer activity, e.g. work 10.1007/s00044-017-1887-9 and others will be a valuable part of this review. There are also missing new group of aminochalcones with great anticancer, e.g. 10.3390/molecules24224129 and 10.1007/s00044-004-0122-7 and others.

Response2:the following text was introduce ,, Objectives for synthesis of new biologically active chalcones are identification of compounds with superior physico-chemical and biological properties. To obtain chalcones with superior anticancer properties, three methods of modulation of natural chalcones were used 1) modulation of two aromatic residues (aldehyde and acetophenone) of chalcones, 2) replacement of aromatic residues with heteroaromatic residues and 3) obtaining hybrids by conjugation with other molecules with antitumor properties. Different substituents on the two aromatic residues of chalcones, depending on their position, influence anticancer capacity by interfering with different biological targets. [220] It is known that biological properties of chalcones are dependent on presence and number of hydroxy and methoxy groups on two aromatic subunits. For example, chalcones with three methoxy groups in molecule on 3, 4 and 5 positions of acetophenone inhibit transport activity of P-glycoprotein and prevent onset of resistance to therapy. [221,157]

XN acyl derivatives

Starting from the idea that esterification of flavonoids is a way to modify hydrophobic character of compounds, a series of mono- and diacetylated derivatives of XN (compounds 14-20, Table I) were synthesized by Zołnierczyk et al. Antiproliferative activities of XN and its derivatives were tested in vitro on HT-29 cell lines. Three compounds from series (compounds 14-16) show XN-like bioactivities and four tested compounds (compunds 17-20) have lower bioactivities. From obtained series, no compound has higher activity than XN. [222]

Another series of XN derivatives was obtained by cyclizing prenyl group from its structure. Thus, a series of six cyclic derivatives of chalcones (compounds 21-26)  were obtained by PopÅ‚oÅ„ski et al. Antiproliferative activity of obtained compounds was evaluated on three human cell lines (MCF-7, PC-3 and HT-27). Potency of XN derivatives was evaluated by SRB method. All compounds obtained show moderate/increased bioactivity, the most vulnerable cell line being MCF-7. Compounds 21 and 23 ((E)-1-(5-hydroxy-7-methoxy-2,2-dimethyl-2H-chromen-6-yl)-3-(4-hydroxyphenyl)prop-2-en-1-one and  (E)-1-(5-hydroxy-7-methoxy-2,2-dimethylchroman-8-yl)-3-(4-hydroxy phe-nyl) prop-2-en-1-one) show the best activity on PC-3 lines, their action being comparable to the activity of the standard (cis-platinum). [223]

Chalcones with nitrogen in molecule

Aminochalcones are known to have strong cytotoxic effects. For example, 2-aminochalcones with a methylenedioxy residue in molecule show very good activity on human nasopharyngeal squamous cell carcinoma (KB-VIN) cell lines. Also, another study indicates that unsubstituted 2-aminochalcones on aldehyde have proapoptotic effects on 20 apoptotic markers. [228]

Starting from fact that different substituted 2-aminochalcones show cytotoxic activity on different cell lines, such as KB (nasopharyngeal squamous cell carcinoma), MCF-7, A-549, 1A9 (ovarian cancer) and are inducers of apoptosis on HT-29 cells, a series of aminchalcone derivatives was obtained (compounds 63-80). Anticancer activity of obtained compounds was evaluated on four cell lines (HT-29, LS180-intestinal human colon adenocarcinoma cell line), LoVo (colon cancer cell line) and LoVo/Dx, by SRB method. Standards used were cis-platine and doxorubicin. Among compounds obtained, the best inhibitory capacity has unsubstituted 2-aminochalcone on aldehyde (compound 63). Activity of compound on HT-29 cell lines is IC50=1.43 μg/ml, being 12 times higher than the activity of cis-platinum (IC50=16.73 μg/ml)  and 4 times lower than activity of doxorubicin (IC50=0.33 μg/ml). From 4-aminochalcones (compounds 75-80), unsubstituted compound on aldehyde (compound75) has the best activity.In the same sense varies activity of 3-aminochalcones (compounds 69-74) on tested cell lines (IC50=1.60-2.13 μg/ml). Potency of these compounds being superior to cis-platine. In case of aminocarboxylic derivatives (compounds 65, 71, 77), position of amino group has a significant impact on IC50. Activity varies in order of 2-amino (compound  65) >3amino (compound 71) >4-amino (compound 77). It has also been observed that incorporation of a nitro group at position 4 of aldehyde (compounds 66, 72, 78) causes a decrease inactivity. [229]

Series of aminochalcones and nitrochalcones were obtained in order to evaluate their cytotoxicity. Activity was determined by MTT method on melanoma cell lines. Compared to nitrochalcones, aminochalcones (compounds 81-91) have advantage of increased solubility in biological media. It was determined that substitution of chalcones with an amino group is favorable, activity of these compounds being superior to nitrochalcones. IC50 values show that presence of amino group on aldehyde causes an increase in cytotoxicity and amino group compounds on acetophenone have a weaker activity. For example, compound 87 (in which amino group is on aldehyde residue) has a higher cytotoxicity than compound 86 (in which amino group is on acetophenone residue). Also, number of methoxy groups on acetophenone determines inhibitory potency of these compounds. Data obtained indicate that aminochalcones substituted with two or three methoxy groups are more active. In case of chalcones substituted with an amino group on 3 position of aldehyde (compounds 87 and 90), cytotoxicity is higher compared to amino-substituted compounds on 4 position (compounds 88 and 89). From chalcones obtained, compound 87 (with an amino group on 3 position of acetophenone and with four methoxy groups) has the best activity. [230]

Wang et al. obtained a series of aminochalcones (compounds 92-103, Table I) that were evaluated for anticancer activity on cell lines (HTC116 and HepG2) by MTT method. All compounds have a good/moderate cytotoxic capacity. Unsubstituted nitrogen compound (compound 92) has the best activity (IC50=0.28±0.06 for HCT116 and 0.19±0.04 for HepG2). Substitution of amine with alkyl groups (compounds 93, 94,96 and 98) causes a significant decrease in antiproliferative activity. A marked decrease in activity was observed on aminochalcone with two 4-(tert-butyl)benzyl residues (compound  99). Results obtained for in vitro evaluation of tubulin inhibition capacity for compound 92 indicate that its molecular target is tubulin, IC50 values for aminocalcone being 7.1 μM, and for colchicine 9.0μM. It has also been observed that aminochalcone (compound 92) has ability to increase proportion of cells in G2/M phase and to block cell cycle. Docking studies for compound 92 show that it binds to binding site of colchicine in tubulin. Aminochalcone adopts an ''L-shaped'' conformation in tubulin pocket. 4-Methoxynaphthyl group of aminochalcone is located in hydrophobic pocket, being surrounded by residues Cys241, Leu248, Ala250, Leu255, Ile318 and Ala354, with which it forms a strong hydrophobic bond. [231]

Modification of amino group in structure of aminochalcones determines every time an increase anticancer activity of compounds. [232] Starting from this premise, we performed a literature study for antitumor activity of some heterocyclic chalcones with a nitrogen in molecule (azoles).

Comment3 I also propose to make subsections for individual compounds, instead of putting all of them together.

Response3:  A more structured text was added.

Comment4 Please review the citation record - should there be any spaces between the dot and the citation brackets.

Response4: citation record was reviewed

Comment5 Please unify all the formulas so that the bonds have the right angles and meet the requirements of IJMS MDPI.

Response5: formulas were corrected

Comment6 Supplementary Materials file missing!

Response6: supplemental  matt was added

Comment7 Figure 2: Incorrect substrate - missing 2'-OH group. Please correct

Response7 figure was corrected

Comment8 Figure 5: "Metochalcone" ??

Response8: corrected

Comment9 Lines 215-219: why is one synthesis example given as general? This is not true, please correct it.

Response9 : corrected

Comment10 Lines 245-260: should this be in the introduction for sure?

Response10: text was rewritten

Comment11 Line 265: it should be "NF-κB"

Response11: corrected

Comment12 Whole article: in vitro and in vivo should be italic.

Response12 corrected

Comment13 Whole article: Latin names for plants should be italic.

Response13 corrected

Comment14 Lines 271: should be "chalcones"

Response14 corrected

Comment15 Whole article: there should be a space between a numeric value and a unit.

Response15 corrected

Comment16 Whole article: it should be trans-chalcone.

Response16 corrected

Comment17 Lines 682, 685: there should be π instead of pi

Response17 corrected

Comment18 Lines 711-712: Surprising and unnecessary paragraph. Please either expand it or delete it. The more so that another review is quoted.

Response18: corrected

Comment19 Lines 843-844: please check the spelling

Response19 corrected

Comment20 Please rebuild the summary.

Response20 summary rebuild

All changes are marked in red 

Reviewer 2 Report

The manuscript is an interesting review concerning anticancer activity of natural and synthetic chalcones – one of class of flavonoid compounds.

I have only few minor comments:

line 14: the name “lycocalcone D” is incorrect, please improve it, it should be “licochalcone D”

Figure 1. should be “dihydroflavonols” instead of “dihidroflavonols”, please correct it

line 164: should be “(aldehyde and acetophenone)”

Figure 3. and Figure 7.  bonds lengths are not equal, please correct

Figure 4. Structure of 2-hydroxychalcone is wrong. It should be actually 2’-hydroxychalcone, which cyclization to flavanone. Figure 4 presents trans-chalcone and flavanone, please correct to 2’-hydroxychalcone, which isomerization to flavanone.

line 198: should be “tyrosine kinase” instead of “tirozinkinase”

Figure 6. Signatures under the structures are incorrect. The first structure is benzaldehyde and the next is acetophenone, please improve it. Also here bonds lengths are not equal, please correct.

line 271 and line 959: should be “Chalcones” instead of “Chalcons”

line 291: should be “Licochalcone A” instead of “Licocalcone A”

line 399 and 435: the same mistake, should be “Licochalcone C” and “Licochalcone D”

Author Response

Comment1 ine 14: the name “lycocalcone D” is incorrect, please improve it, it should be “licochalcone D”

Response1: corrected

Comment2 Figure 1. should be “dihydroflavonols” instead of “dihidroflavonols”, please correct it

Response 2 corrected

Comment3 line 164: should be “(aldehyde and acetophenone)”

Response3 corrected

Comment4 Figure 3. and Figure 7.  bonds lengths are not equal, please correct

Response4 corrected

Comment5Figure 4. Structure of 2-hydroxychalcone is wrong. It should be actually 2’-hydroxychalcone, which cyclization to flavanone. Figure 4 presents trans-chalcone and flavanone, please correct to 2’-hydroxychalcone, which isomerization to flavanone.

Response5 corrected

Comment6 line 198: should be “tyrosine kinase” instead of “tirozinkinase”

Response6 corrected

Comment7 Figure 6. Signatures under the structures are incorrect. The first structure is benzaldehyde and the next is acetophenone, please improve it. Also here bonds lengths are not equal, please correct.

Response7 corrected

Comment8 line 271 and line 959: should be “Chalcones” instead of “Chalcons”

Reponse8 corrected

Comment9 line 291: should be “Licochalcone A” instead of “Licocalcone A”

Response9 corrected

Comment10 line 399 and 435: the same mistake, should be “Licochalcone C” and “Licochalcone D”

Response10 corrected

All changes are marked in red 

Round 2

Reviewer 1 Report

Thank you for all corrections and changes.

Some additional comments:

Figure 4: Wrong substrate!! Please correct. Moreover, there should be "2'-hydroxy-5'-nitrochalcone"

Line 735: should be: "Żołnierczyk"

Lines 806-824: should be "μg/mL" not "μg/ml"

Line 847: should be "trans" (italic)

Line 1753" should be "Żołnierczyk"

Line 1778: should be "Żarowska"

After the changes, the article will be suitable for publication in IJMS MDPI

Author Response

Thank you for reviewing our manuscript

Comment1: Figure 4: Wrong substrate!! Please correct. Moreover, there should be "2'-hydroxy-5'-nitrochalcone"

Response1: Corrected as suggested

Comment2: Line 735: should be: "Żołnierczyk"

Response2: Corrected as suggested

Comment3: Lines 806-824: should be "μg/mL" not "μg/ml"

Response3: Corrected as suggested

Comment4: Line 847: should be "trans" (italic)

Response4: Corrected as suggested

Comment5: Line 1753" should be "Żołnierczyk"

Response5: Corrected as suggested

Comment6: Line 1778: should be "Żarowska"

Response6: Corrected as suggested